# Downsizing of animal communities triggers stronger functional than structural decay in seed-dispersal networks

Isabel Donoso[1 ✉], Marjorie C. Sorensen [1,2,3], Pedro G. Blendinger [4], W. Daniel Kissling [5], Eike Lena Neuschulz[1], Thomas Mueller [1,3] & Matthias Schleuning [1]

Downsizing of animal communities due to defaunation is prevalent in many ecosystems. Yet, we know little about its consequences for ecosystem functions such as seed dispersal. Here, we use eight seed-dispersal networks sampled across the Andes and simulate how downsizing of avian frugivores impacts structural network robustness and seed dispersal. We use a trait-based modeling framework to quantify the consequences of downsizing—relative to random extinctions—for the number of interactions and secondary plant extinctions (as measures of structural robustness) and for long-distance seed dispersal (as a measure of ecosystem function). We find that downsizing leads to stronger functional than structural losses. For instance, 10% size-structured loss of bird species results in almost 40% decline of long-distance seed dispersal, but in less than 10% of structural loss. Our simulations reveal that measures of the structural robustness of ecological networks underestimate the consequences of animal extinction and downsizing for ecosystem functioning.

[1] Senckenberg Biodiversity and Climate Research Centre (SBiK-F), Senckenberganlage 25, 60325 Frankfurt am Main, Germany. [2] Department of Integrative Biology, University of Guelph, Guelph, 50 Stone Road East, N1G 2W1 Ontario, Canada. [3] Department of Biological Sciences, Goethe University Frankfurt, Max-von-Laue-Str. 9, 60438 Frankfurt am Main, Germany. [4] Instituto de Ecología Regional, Consejo Nacional de Investigaciones Científicas y Técnicas (CONICET) - Universidad Nacional de Tucumán, CC 34, 4107 Yerba Buena, Tucumán, Argentina. [5] Institute for Biodiversity and Ecosystem Dynamics (IBED), University of Amsterdam, P.O. Box 94240, 1090 GE Amsterdam, The Netherlands. ✉email: isa.donoso.cuadrado@gmail.com

Species defaunation, i.e., the extinction of animal species due to overhunting and habitat loss, has been identified as a main trigger for the global biodiversity crisis[1,2]. Human overexploitation of animals occurs in all types of biomes[3,4] and is especially prevalent in species-rich tropical ecosystems[5,6]. Defaunation is often size-selective, targeting especially the largest species, which leads to downsizing of animal communities[7,8]. Since biodiversity sustains essential ecosystem functions, such as pollination or seed dispersal by animals[9,10], it is pivotal to understand how biodiversity loss and concurrent ecological downsizing modify ecosystem functioning.

To assess how changes in biodiversity impact ecosystem functioning, recent studies advocate for approaches based on functional traits that assess what species in a community do rather than how many there are[11,12]. Linking the actual contribution of species to ecosystem functions requires a comprehensive framework based on both the number of species and their functional traits as key biotic drivers of ecosystem functioning[13]. Trait-based approaches consider that species differ in their extinction proneness depending on their vulnerability to disturbance (i.e., determined by their response traits), but also in their contribution to ecosystem functions (i.e., determined by their effect traits)[14,15]. Understanding whether the most vulnerable species are also the species that contribute most to ecosystem functioning remains a significant challenge, in particular for real-world ecosystems.

In ecological communities, species are organized in complex interaction networks[16,17]. Interactions between species in these networks are often influenced by the traits of the interacting species[18,19]. In particular, size determines which species interact in ecological networks[3,20]. Likewise, size matching has been described as a crucial mechanism mediating defaunation effects on ecosystem functioning[21,22]. Yet, studies analyzing the consequences of species loss from ecological networks primarily hinge upon decreases in the structural robustness of ecological communities. Decreases in the structural robustness are usually measured using network metrics, such as changes in interaction strength[3,23] or secondary extinctions as a consequence of species loss from the other trophic level[17,24,25]. Measures of structural robustness, however, only reflect species' contributions to the number of interactions, ignoring that species are not equivalent in terms of their functional contribution[26]. Although a few previous studies have analyzed the consequences of defaunation for certain ecosystem functions[1,27,28], it remains unclear to what extent structural changes can represent functional changes in ecological networks.

A pivotal ecosystem function is animal-mediated seed dispersal[16], especially in tropical forests where about 70% of plant species require birds to disperse their seeds[29]. Mutualistic networks composed of fleshy-fruited plants and frugivorous birds are essential to maintain the structure and diversity of tropical plant communities. In particular, long-distance seed dispersal (LDD) by avian frugivores shapes the spatial scale of plant demographic processes[30] and promotes genetic exchange and connectivity in fragmented landscapes in both temperate and tropical ecosystems[22,31]. Defaunation is expected to lead to changes in seed dispersal by animals[7], and changes in LDD could directly affect the spatial distribution of plants and their potential to colonize new localities in response to global change[32,33]. To estimate such functional consequences of defaunation, seed-dispersal distances have to be estimated for entire plant communities[33]. So far, research efforts on estimating seed-dispersal distances have mainly focused on specific plant species[7,34] or simulated seed dispersal only for small communities[35,36].

Here, we use a simulation approach that integrates information from empirical interaction networks and movement ecology with a trait-based modeling framework to estimate total dispersal distances across all plant species in a community[37] (Supplementary Fig. 1; Methods). We apply this model to empirical data from eight seed-dispersal networks collected across the tropical Andes including 11,572 interactions between 227 plant and 179 bird species (see Methods and Supplementary Table 1 for details). We then implement and compare two animal extinction scenarios to the modeling framework; one random and one deterministic scenario removing bird species from the largest to the smallest species. This allows us to predict the consequences of random and size-structured defaunation for entire plant communities (Fig. 1a). First, we evaluate changes in structural robustness defined as i) the number of lost interactions (Fig. 1b) and ii) the secondary extinction of plant species (Fig. 1c). Second, we quantify functional changes by analyzing how LDD, defined by the 0.95 quantile of community-wide seed-dispersal distances, varies in response to downsizing (Fig. 1d). We compare to what extent changes in structural robustness in interaction networks differ from functional changes in response to downsizing of animal communities. We find evidence that the consequences of downsizing are more severe for seed dispersal than for structural robustness, indicating that measures of functional changes are required to capture the ecosystem consequences of animal loss.

## Results

**Structural changes in response to animal extinction**. We simulated changes in structural robustness (in terms of interaction loss and secondary plant extinctions) in response to bird species loss using two extinction scenarios representing (1) a random extinction sequence, and (2) a deterministic extinction sequence according to body mass (i.e., defaunation-driven downsizing). Changes in structural robustness along the extinction sequences were estimated relative to the original network including all species. Across the eight empirical networks, the number of interactions declined linearly in response to size-structured extinction (Fig. 2a). Differences between deterministic and random extinctions were weak and only larger for the deterministic extinction sequence after about 75% of bird species had been removed. Further, we found that the number of secondarily extinct plant species followed an accelerating curve in response to bird species loss in both scenarios (Fig. 2b). After removing 10% of the bird species, the downsizing scenario showed more secondary plant extinctions than the random scenario (Fig. 2b). Across the eight networks, removal of 50% of bird species resulted in about 50% of lost interactions and about 40% of secondary plant extinctions (Fig. 2b). The eight individual networks showed similar patterns of interaction loss and consistent trends of secondary plant extinction across the deterministic bird extinction sequence (Supplementary Fig. 2).

**Functional changes in response to animal extinction**. We evaluated functional changes along the two simulated extinction sequences with changes in LDD of the entire plant community, defined as the 0.95 quantile of community-wide seed-dispersal distance (see Supplementary Fig. 1 and Methods section). Downsizing of animal communities led to strong functional losses in terms of LDD relative to LDD in the original network. Across the eight empirical networks, we found a large difference between the downsizing and the random extinction sequence (Fig. 3). While size-structured defaunation led to rapid functional decays following a sigmoidal curve, the random removal sequence only led to a sharp decline of LDD after about 90% of the bird species were removed. Trends were qualitatively similar across the eight Andean networks (Supplementary Fig. 3).

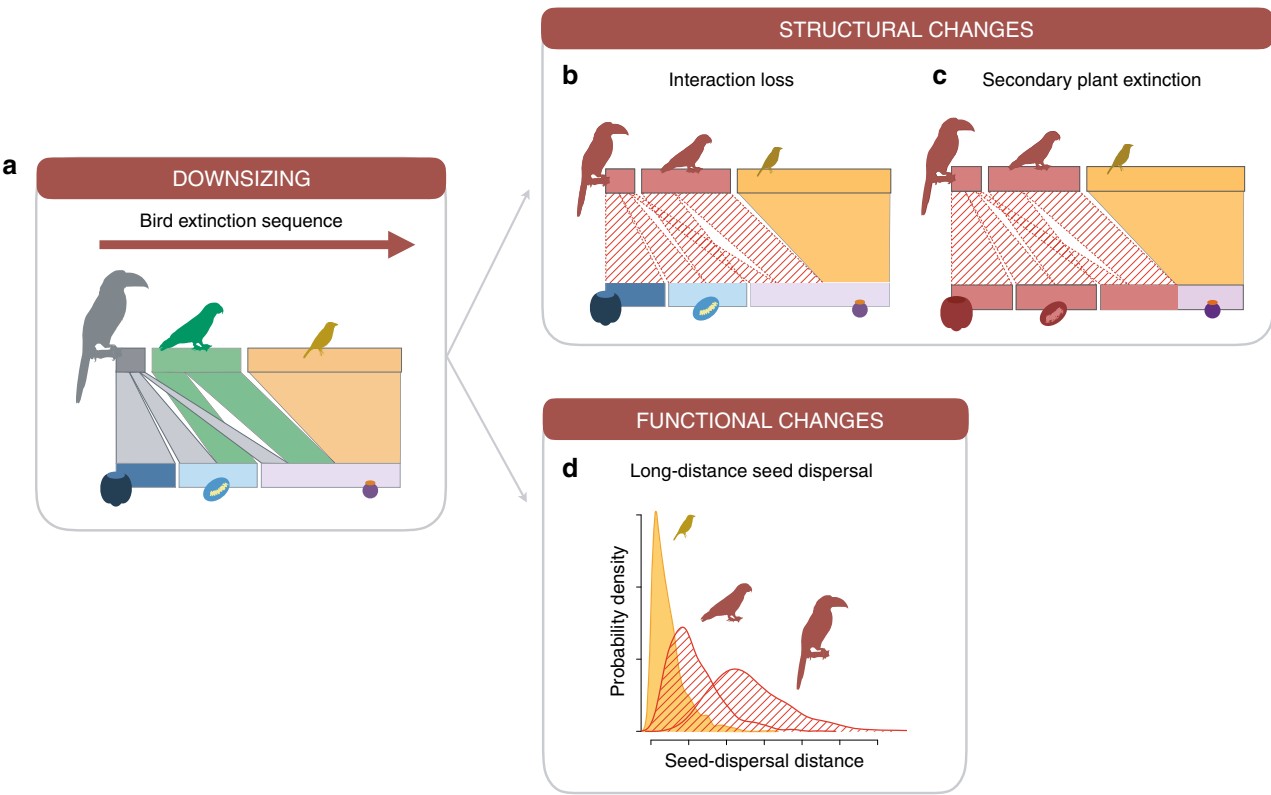

**Fig. 1 Potential structural and functional consequences of ecological downsizing. a** Size-dependent extinction of frugivorous birds results in a community downsizing which translates into structural changes, in terms of **b** interaction loss and **c** secondary plant extinctions, and **d** functional changes in terms of long-distance dispersal. Bird species lost from the community are represented by red silhouettes. Changes in structural robustness are quantified in terms of the number of lost interactions (indicated by red stripes) and number of plant species that became secondarily extinct (depicted by red silhouettes). Functional changes are illustrated by assessing how the loss of animal species affects seed-dispersal distances, depicted by a probability distribution of dispersal distances (red stripes indicate the lost parts of the seed-dispersal kernel). Bird silhouettes by I. Donoso.

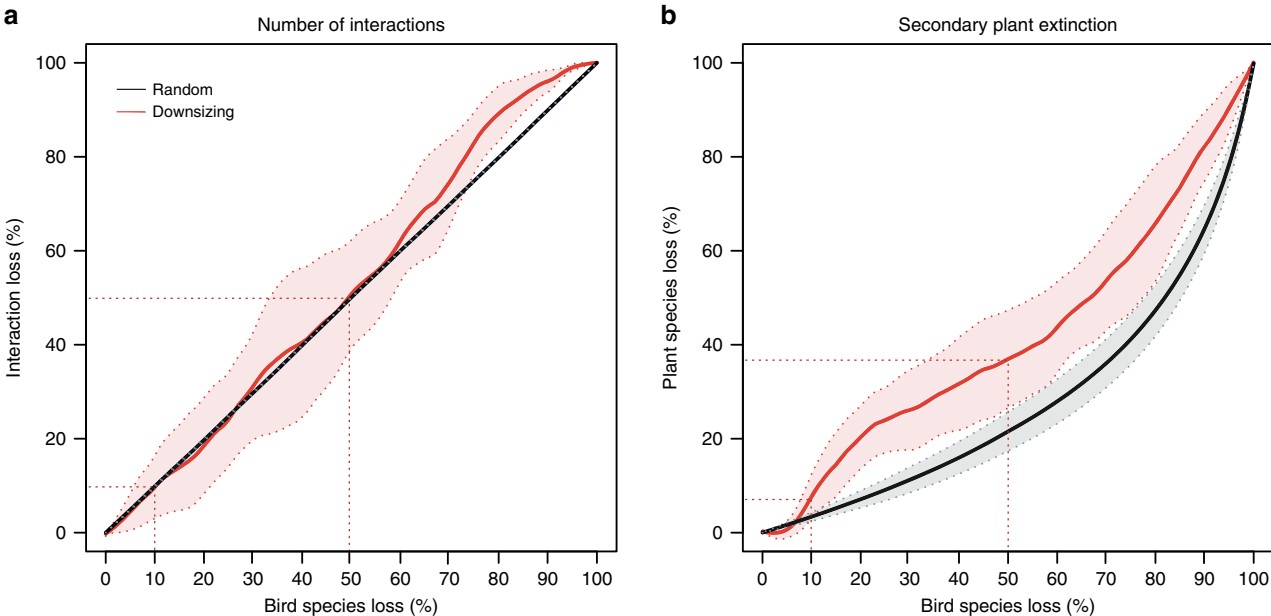

**Fig. 2 Structural changes in seed-dispersal networks in response to simulated animal extinctions.** The consequences of size-structured bird extinctions (downsizing; red line) and random bird extinctions (mean across 1000 iterations; black line) are shown for **a** the number of interactions and **b** secondary plant species extinctions. Values on the y-axis represent the proportional loss relative to the number of interactions and plant species in the original network. Red and gray areas represent the 95% confidence intervals across the eight Andean seed-dispersal networks. Red-dashed lines in **a** and **b** indicate the percentage of structural loss after 10 and 50% of bird species had been removed in the downsizing scenario.

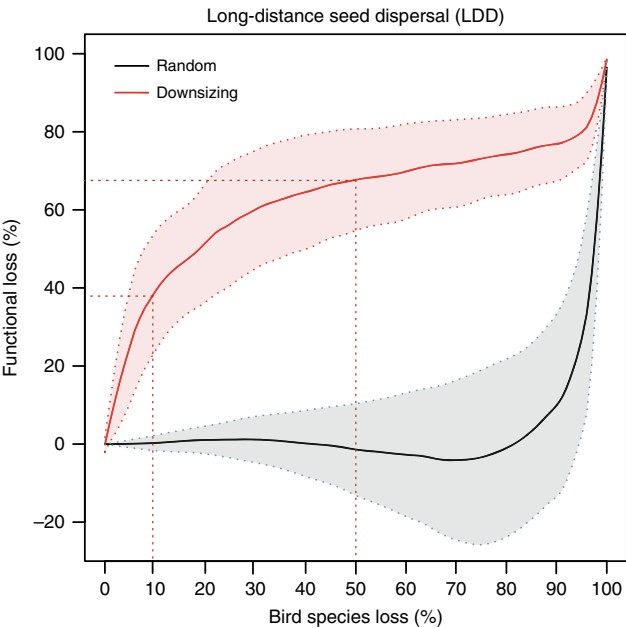

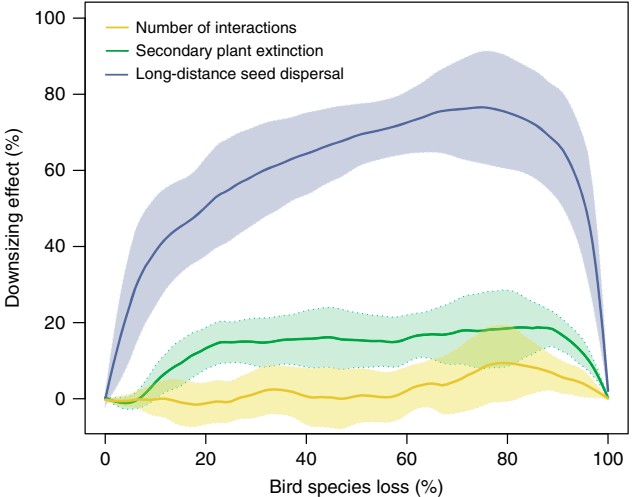

**Fig. 3 Functional changes in seed-dispersal networks in response to simulated animal extinctions.** The consequences of size-structured bird extinctions (downsizing; red line) and random bird extinctions (mean across 1000 iterations; black line) for long-distance seed dispersal (LDD) are shown. Values on the y-axis show the percentage of functional change defined by the 0.95 quantile of the community-wide seed-dispersal distance relative to the value in the original network. Red and gray areas represent the 95% confidence intervals across the eight Andean seed-dispersal networks. Note that the confidence intervals overlap with zero along the whole random extinction sequence. Red-dashed lines indicate the percentage of functional changes after 10 and 50% of bird species had been removed in the downsizing scenario.

**Downsizing triggers stronger functional than structural losses**. Overall, the simulations of avian downsizing showed more pronounced changes in LDD than in both measures of structural robustness (Fig. 4). For example, the size-structured removal of 10% of bird species resulted in less than 10% of structural losses in terms of the number of interactions and secondary plant extinctions (Fig. 2b), but an estimated functional loss in LDD of almost 40% (Fig. 3). The effect size of downsizing, defined by the difference in the proportional losses between downsizing and the random extinction scenarios, was more than three times larger for functional than for structural losses along the entire extinction sequence (Fig. 4).

## Discussion
We assess the structural and functional consequences of ecological downsizing for entire plant communities dispersed by avian frugivores. Our simulations, based on empirical plant-frugivore networks, reveal that functional effects (in terms of long-distance seed dispersal) are stronger than effects on structural robustness (in terms of the number of interactions and secondary extinctions). Our simulations show that measures of the structural robustness of ecological networks are likely to underestimate the functional consequences of animal downsizing for ecosystems.

Size-structured extinction of bird species had a disproportionate impact on seed dispersal, reducing LDD substantially after the loss of a few avian dispersers. The loss of large bird species might cause a strong reduction in LDD because small frugivores are not able to compensate for long-distance dispersal

**Fig. 4 Effect size of downsizing on structural and functional losses in seed-dispersal networks.** The effect of downsizing is defined as the difference in the proportional losses between the size-structured and random extinction scenarios. Effect sizes for the number of interactions and secondary plant extinctions (structural loss; yellow and green lines, respectively) are compared to those for long-distance seed dispersal (functional loss; blue line) along the entire extinction sequence. Yellow, green and blue areas represent the 95% confidence intervals across the eight Andean seed-dispersal networks for the effect size of structural and functional losses.

by large-bodied frugivores[34]. Empirical studies have indeed shown that the loss of large frugivores led to a significant reduction of seed-dispersal distances for individual plant species[7]. Such effects could be further exacerbated because species abundance usually declines sharply before a species completely disappears, a process referred to as functional extinction[26]. From a community-wide perspective, reductions in LDD might be further amplified because small-seeded plants gain in importance relative to large-fruited species that are dispersed over longer distances (as previously shown in refs. [19,21]). Our findings suggest that bird species with different body sizes strongly differ in their functional roles, e.g. because of differences in their gut passage time, their home range or differences in space use[38,39]. Consistent with our finding, previous studies have proposed functional complementarity as a general mechanism underpinning the link between biodiversity and ecosystem functioning for pollination[40] and other functions derived from trophic relationships[41,42].

In contrast to LDD, we found weak differences between random and deterministic simulated extinction sequences for both metrics of structural robustness. Along the sequences of bird extinction, interaction loss increased almost linearly, suggesting that different bird species contributed similar quantities of interaction events, independent of avian body mass. In contrast to our finding, it has been shown that the amount of dispersed seed can increase with avian body mass, especially if large frugivores are abundant[43] (see also network Colombia 1 in Supplementary Fig. 2). However, body mass is not universally related to the number of interactions, consistent with a recent meta-analysis across a large number of Neotropical and Afrotropical seed-dispersal networks[44]. Avian body mass, therefore, does not seem to be a generic effect trait that determines the quantitative contribution of frugivores to seed dispersal. In terms of the structural robustness of seed-dispersal networks, we further found that simulated secondary extinctions of plant species occurred at a lower rate than bird species loss. This is consistent with a

previous coextinction simulation of seed-dispersal networks[45] and is also reflected by the rarity of documented secondary extinctions in plant-frugivore communities[46,47]. Our simulations further show that downsizing had slightly stronger effects on secondary plant extinctions than random species loss, suggesting that large frugivores play a unique role for the dispersal of some plants[25]. Nevertheless, from a community-wide perspective, secondary extinctions are unlikely, given the resource overlap among frugivores[16,48] and the ability of frugivores to switch between fruit resources[49]. For instance, previous studies have shown that small frugivores, such as tanagers, can buffer secondary extinctions after the loss of large seed dispersers[19,21]. Structural redundancy has also been shown for other types of ecological networks, such as plant-pollinator, plant-herbivore, and host-parasitoid systems[40,41,50]. Consequently, the structural robustness of ecological networks appears to be high in response to the loss of individual species[51].

The contrasting results for structural and functional losses might be explained by differences in the relationship between extinction proneness and species' structural and functional contributions to the community (i.e., the correlation between response and effect traits[13]). Redundancy among frugivores has been proposed in cases where species that are more susceptible to extinction play a structural role that could also be fulfilled by other species in the community[11]. On the other hand, the rapid functional decay, revealed by our simulations, is expected if extinction-prone species contribute unique functions to the ecosystem[11,15]. Recent studies have further suggested that ecosystem functions appear to be more vulnerable to biodiversity loss if trait variation in a community is high[15]. Therefore, the application of trait-based approaches in network ecology is a promising way forward to assess the impacts of global change on ecosystem functioning.

Here, we assess the structural and functional changes in empirical seed-dispersal networks after simulating species extinctions as a function of avian body size. This approach is justified because previous studies have shown that extinctions in both aquatic and terrestrial systems occur more frequently in large-bodied organisms (see ref. [52] and references therein). Nevertheless, using only body size as a response trait simplifies the complexity of real-world ecosystems and may not be sufficient to quantify the extinction risk for all taxa[33]. For instance, specialization on host plants increases the extinction risk in butterflies[53]. Future studies could therefore expand our results by also considering other response traits, such as resource and habitat specialization[54] or population size and generation time[55]. On the other hand, consequences of species loss could be mitigated by compensatory mechanisms of persisting species, such as density compensation[56]. In our simulation, we assumed no density compensation by the remaining species in the community which is consistent with empirical evidence[57,58]. Our simulations also did not account for potential behavioral differences among bird species unrelated to body size (but see Supplementary Note 1 for simulation results excluding potential regurgitation events). Future studies could therefore aim at including additional traits related to habitat use and behavior of animal seed dispersers, such as wingspan[59] or stratum of movement[60].

Using empirical network data and extinction simulations, we show how the loss of a few, large-bodied seed dispersers substantially lowers community-wide LDD despite a high degree of structural redundancy in seed-dispersal networks. We conclude that network studies that focus only on the structural robustness of networks are prone to underestimate the functional consequences of animal loss. The consequences of animal downsizing for ecosystems therefore reach far beyond the emblematic examples of apex predator loss[61,62] and can modify ecosystems even if only a small fraction of animal species is lost from diverse ecological communities.

## Methods

**Seed-dispersal networks**. We compiled data of weighted interaction networks between fleshy-fruited plants and frugivorous birds which were collected at eight different locations across the tropical Andes[19]. The eight networks included two networks from Colombia, two networks from Ecuador, two networks from Peru, one network from Bolivia, and one network from Argentina, covering an elevational range from 1,000 to 3,000 m a.s.l. (see Supplementary Table 1 and ref. [63]). Sampling effort was high in all networks (range of sampling hours: 300–960, mean = 606, standard deviation = 224), and all networks were collected in near-natural, montane forests[19]. Overall, we used data collected during more than 4800 sampling hours, during which 11,572 events of fruit consumption were recorded. Networks comprised 179 bird species interacting with 227 plant species. For all bird species, we recorded data on avian body mass from the EltonTraits database[64].

**Community-wide seed-dispersal distance**. Functional changes in seed-dispersal networks were evaluated in terms of changes in LDD, defined by the 0.95 quantile of the seed-dispersal distance of all plant species in a community[65]. To estimate seed-dispersal distances from a community-wide perspective, we applied a method developed by Sorensen et al.[37], which combines information from empirical interaction networks with a trait-based mechanistic seed-dispersal model based on animal body size (Supplementary Fig. 1).

Data from each empirical interaction network informed the seed-dispersal model about the relative contribution of each avian disperser to the total number of seed-dispersal events (Supplementary Fig. 1a). Since only a few interaction events had been observed for some species pairs, we multiplied the number of interaction events in each empirical network by 1000 to reduce stochastic noise in the seed-dispersal models. That is, we simulated more interaction events than had been empirically observed, but maintained the relative frequency of interaction links between plant and bird species from the empirical observations. To estimate the seed-dispersal distance resulting from each plant-bird interaction event, we applied a trait-based mechanistic seed-dispersal model based on allometric relationships of body mass (BM) with both gut passage times (GPT) and movement distance, i.e., the two main components determining seed-dispersal distances[34,65,66] (Supplementary Fig. 1b).

Mean GPT of each bird species was estimated by fitting an allometric equation between GPT and BM with the most up to date empirical information collected from the literature[37]:

$$GPT(h) = 4.5BM[kg]^{0.5} \qquad (1)$$

This allometric equation expanded the one proposed by Robbins et al.[67] by increasing sample size and including only frugivorous birds ($n = 37$, $R^2 = 0.69$, $p < 0.001$). Data on GPT of birds foraging in natural environment was available from feeding trials for 34 bird species[37], which represented the full range of body mass variation recorded in the Andean communities (Supplementary Fig. 4; see also Supplementary Data 1 for detailed information about the species recorded in the Andes). In our simulations, we focus on dispersal events resulting from endozoochory via defecation, although seeds might also be dispersed via regurgitation, which generally results in shorter seed-dispersal distances[68]. When excluding potential regurgitation events according to the ratio between fruit and bill size[69,70], simulation results remained qualitatively identical (see Supplementary Note 1, and Supplementary Figs. 7–9).

Because a general relationship between BM and home range size is unknown for birds and because bird species often make movements beyond their home range, we used flight speed (FS) as a proxy to calculate movement distance[71,72]. Mean FS of each bird species was estimated by applying the allometric equation between FS (in no-wind conditions) and BM developed by Tucker et al.[73]:

$$FS(m\ s^{-1}) = 15.7BM[kg]^{0.17} \qquad (2)$$

Equation (2) derives flight speed from avian aerodynamics measures collected during wind tunnel experiments, which makes it broadly applicable to bird species[73]. According to Sorensen et al.[37], slight variation in the flight speed parameters was of little relevance for the estimates of LDD.

To simulate avian seed dispersal, we applied the two allometric equations to each interaction event between a bird and a plant species in the networks. To this end, we drew the value of the GPT for the ingested seed and the FS of the bird species from their respective probability distributions (Supplementary Fig. 1b), which were multiplied to assess the displacement distribution (Supplementary Fig. 1c). GPT was selected from a Gamma probability distribution[74] because it most closely matches the GPT data found in empirical studies[65,74]. The Gamma distribution was parametrized using the shape ($k$) and scale ($\theta$) parameters, which can be defined in terms of the empirical mean GPT calculated for the particular bird species involved in the interaction ($\bar{t}$) and the variance ($s^2$) as follows:

$$k = \bar{t}^2/s^2 \qquad (3)$$

$$\theta = s^2/\bar{t} \qquad (4)$$

where $(\bar{t})$ is the mean GPT estimated in Eq. (1) and $(s^2)$ was the mean variance found in the published studies (see Sorensen et al.[37]). FS was selected from a normal probability distribution[75], which was parameterized by using the empirical mean FS estimated from allometric Eq. (2) and the average standard deviation of flight speed reported for bird species in Alerstam et al.[76]. Bird species larger than 1770 g were excluded from this calculation since none of the species in the Andean interaction networks exceeded that body size.

We estimated seed-dispersal distance by multiplying GPT and FS (the values drawn from their respective probability distributions). To account for time spent resting and non-directional movements of birds, the estimated dispersal distance for each event was multiplied by a calibration term, which was estimated following Schurr et al.[71] by combining Eqs. (1) and (2) (with hours converted to seconds):

$$z = fc15.7(16{,}200)\text{BM}^{0.17+0.5} \text{ standard units of } z[\text{m}] \text{ and } \text{BM}[\text{kg}] \quad (5)$$

where $z$ is seed-dispersal distance; $f$ is time allocated to movement as a constant fraction of the GPT; and $c$ is a straightness factor accounting for movement deviating from a straight line ($c$ is 1 if movement occurs in a straight line). Equation (5) was compared to field-based seed-dispersal studies that reported empirical dispersal distances[37]. The calibration term (defined by the product of $f$ and $c$) was calculated by computing the ratio between the allometric constant from Eq. (5) and that derived from the empirical studies. The analytically derived calibration term is $fc = 0.002$ (see ref. [37]).

Simulations of seed-dispersal events across all observed avian seed dispersers resulted in seed-dispersal distance kernels for each individual plant species in the respective empirical network (Supplementary Fig. 1c). To estimate the distribution of seed-dispersal distances for the entire plant community (i.e. community-wide seed-dispersal distance), we combined the dispersal kernels across all plant species within each interaction network (Supplementary Fig. 1d). Following Perez-Méndez et al.[7] and Pires et al.[65], we defined the 0.95 quantile of the displacement distance as an approximation of long-distance seed-dispersal events (see Supplementary Fig. 5).

**Trait-based extinction model to simulate species defaunation**. We simulated how downsizing of animal communities triggers structural and functional changes in seed-dispersal networks by implementing an extinction model based on animal size[21]. To this end, we evaluated structural and functional changes under a scenario representing a deterministic extinction sequence based on avian body mass (i.e. removing birds from the largest to the smallest species), and compared the results to those derived from a scenario representing a random sequence of bird extinction (defined by the mean of 1000 random sequences of bird extinction). For each of the eight Andean networks and extinction sequence, we calculated functional and structural changes corresponding to each value of bird richness by scaling these values relative to the value in the original, full assemblage. With this approach, we obtained the percentage of structural and functional loss or gain compared to the values from the original assemblage, which enabled us to directly compare structural and functional changes in the networks. We generalized the results across the eight communities by fitting a generalized additive model on the relationship between the percentage of bird species loss and (i) interaction loss, and (ii) secondary plant extinction (structural changes), and (iii) changes in LDD (functional change), by using the function loess in R[77]. The span parameter (i.e. the parameter controlling how the fit at a specific point in the series weights the data nearest to it) was set to the minimum value of each network to visually resemble the empirical data and avoid over-fitting. For each generalized additive model, we derived 100 predicted values of structural and functional changes per network along each of the two extinction sequences. We estimated the structural and functional changes across the eight communities by calculating the mean and confidence intervals of the predicted values across networks for each extinction sequence. Finally, to compare the effect of downsizing on structural and functional changes, we calculated an effect size by computing the difference in percentage loss between downsizing and random extinctions for structural and functional changes along the entire extinction sequence for each network. To generalize effect sizes across networks, we fitted generalized additive models, as described above.

**Reporting summary**. Further information on research design is available in the Nature Research Reporting Summary linked to this article.

## Data availability
Original data of the eight seed-dispersal networks are stored and metadata are available at the Senckenberg Metacat Data Repository: https://doi.org/10.12761/SGN.2018.10237 (ref. [63]). Data on avian body mass was recorded from the EltonTraits database (ref. [64], available at http://www.esapubs.org/archive/ecol/E095/178/).

## Code availability
The R code used to estimate community-wide seed-dispersal distance is available as Supplementary Material of the preprint stored at https://doi.org/10.1101/2020.02.23.958454. The R Source code of the functions to estimate structural and functional changes under the two extinction scenarios are provided as Supplementary Material (Supplementary Code 1).

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

## Acknowledgements

We thank Jörg Albrecht for his suggestions on data analysis. We are also grateful to Irene Bender, Matthias Dehling, Marcia Muñoz, Marta Quitián, Francisco Saavedra, Vinicio Santillán and the members of the Laboratorio de Ecología de Aves (IER, Argentina) for recording data on interaction networks. I.D. and M.C.S. were funded by the Alexander von Humboldt Foundation. E.L.N. and M.S. obtained funding by the German Research Foundation (DFG) in the framework of the Research Bundle 823–825 "Platform for Biodiversity and Ecosystem Monitoring and Research in South Ecuador" (PAK 825/1) and the Research Unit FOR2730 "Environmental changes in biodiversity hotspot ecosystems of South Ecuador: RESPonse and feedback effECTs". T.M. was funded by the Robert Bosch Foundation. The study in Argentina was funded by CONICET (PIP 2009-1025 and 2014-592) and ANPCyT (PICT 2013-1280) granted to P.G.B. W.D.K. acknowledges financial support from the Netherlands Organization for Scientific Research (824.15.007) and the University of Amsterdam Faculty Research Cluster 'Global Ecology'.

## Author contributions

I.D. and M.S. conceived the study. M.C.S. and T.M. developed the approach to estimate community-wide seed-dispersal distance. I.D. processed and analyzed the data and wrote the first draft of the manuscript with input from M.S. M.C.S., P.G.B., W.D.K., E.L.N., T.M. and M.S. revised the manuscript and contributed to the final version.

## Competing interests

The authors declare no competing interests.
