## [Peer Review File · Nature Communications]

Reviewers' Comments:

Reviewer #1:

Remarks to the Author:

This paper compares commonly used simulations of coextinction within mutualistic networks to simulations of the loss of ecological function (long-distance dispersal) following simulated extinction. The key conclusions – that functional loss can outpace species loss and that ecological function can be better incorporated in network models – are important and I think articulated well. From one perspective, the paper's primary conclusions mirror basic ideas/predictions present in the literature, and the qualitative conclusions of the model are more or less guaranteed to be found on first principle. Because networks are typically nested and larger animals can eat a broader diversity of fruits, the loss of the largest animals will leave disproportionately many plant species without partners. Because dispersal distances increase with body size, and because body size distributions are roughly log-normally distributed, the loss of the largest animals is essentially guaranteed to cause disproportionate decreases in dispersal distance.

All of that said, the integration of field network data and trait data, as well as trait-based estimates of dispersal distances, allows the authors to get a quantitative estimate of the scope of these simulated downsizing effects if they were to occur in a real community, making this a useful contribution to the field. I also think it is a useful strategy to consider functional components within the context of the commonly used topological coextinction models. Hopefully this makes it easier to put these functional loss ideas in the context of previous literature using such coextinction simulations. So overall I think the conclusions of the model, and the large scope of functional loss demonstrated, provides a useful demonstration applicable to those modeling mutualistic networks and those studying them in the field. I generally found the presentation to be clear. My one major comment points out a methodological question that may quantitatively impact the conclusions.

Major comments

Displacement values are simulated by taking the product of a draw from a GPT distribution and a draw from the FS distribution. This product seems to be how displacement values are typically simulated. What I do not understand is why this product is then used as the mean of a lognormal displacement distribution, with the actual displacement value drawn from that distribution (L 318). Ecologically, I do not understand where this additional variation is coming from. Why I think this is important is because the long tail of the lognormal distribution will make some very high values for displacement and, in turn, dispersal distances. If I understand correctly, these will be disproportionately long for large bodied animals, and this could bias upwards the estimates of functional loss caused by large frugivore loss (which are not measured on the log scale). The authors should avoid this additional step or justify the biological meaning of this additional distribution. I suspect the functional downsizing effects would be less pronounced without this additional step.

I also suggest clarifying how functional loss is actually calculated. I think it's the 95th percentile of the simulated dispersal distance of whatever frugivores persist versus the 95th percentile of simulated dispersal distance of the full frugivore assemblage before any extinction, but this was unclear in the text.

Specific comments

L25. How the term "structural stability" is used here is different from how structural stability is typically used, which is in the context of dynamical models (e.g., Rohr et al. 2014 Science). Perhaps robustness is a more appropriate term?

L52. Rephrase "size has been described as a crucial mechanism mediating defaunation effects on ecosystem functioning". Size itself is not a mechanism.

L166. This reads as a very general and strong conclusion, so it is worth being clear that these are simulated extinctions.

L170. Maybe a word is missing here - perhaps "compensate against secondary plant extinctions"

L 182. Rephrase "is little limited"

Reviewer #3:

Remarks to the Author:

The authors use extinction simulations in seed dispersal networks to investigate how the loss of frugivorous birds could impact the structure of the network and seed dispersal services. Although the study is interesting and well-written, it is structured in such a way that the authors are continuously comparing the functional versus structural consequences of frugivore loss. Because they only consider that structure changes when species are lost, it is not surprising that structure will change less than function, especially if they consider plants will only die out when all birds that disperse them are extinct. Actually, assuming secondary extinctions of plants due to the loss of frugivores is not realistic. There is very limited evidence that the loss of frugivores result in local extinction of plants, although it can result in dispersal limitation with consequences for population genetics, population size and structure and spatial distribution. I'm not saying that extinction simulations are not informative, but it seems more reasonable to measure the loss of interactions instead of species loss. If a plant loses all its dispersers it becomes disconnected from the network, but it is unlikely to go extinct, especially in short timespans. If the authors consider the number or proportion of lost interactions or had some way to evaluate changes in the distribution of interaction weights, for instance, the results on the structural versus functional consequences of species loss could be very different. That said, I think relying on this comparison is not the best way to motivate the paper. Instead of opposing the two consequences it would be more interesting to ask how changes in topology translate into functional consequences.

Another important issue is that some of the simplifying assumptions used to estimate seed dispersal distances, disregard important behavioral and physiological characteristics of the bird species that have an important impact on their role as seed dispersers (see below)

Detailed comments:

The introduction is a bit tiresome due to excessive repetitions on what is unknown, unclear or little understood.

Figure 1. secondary extinctions of plants due to the loss of frugivores seem rather unlikely. The consequences are probably seen in terms of genetic diversity and changes in spatial distribution over time.

Ln 143. This comparison is strongly dependent on how you determine secondary extinctions. Since you assume plants will only undergo extinction after they lose all their partners, then it is expected that structure will be more robust than function.

Ln 164. The discussion also hinges on the structure vs function comparison, which I think is not the best strategy since it depends on how you measure each

Ln 166-169. These results can be obtained from structural network metrics and redundancy metrics. I don't think this is an important finding emerging from your simulations, but an explanation for the obtained results.

Ln 199. It is not clear if you are referring to your findings or those of the cited references

Ln 272. This decision needs to be explained in more detail

Ln 281. how good is the fit of this allometric relationship?

Ln 287-298. Movement distance varies a lot depending on habitat preferences of different bird species. Understory specialists, such as some of the Passerelidae species in the networks tend to fly very short distances whereas canopy birds with similar body mass, such as many Thraupidae in the dataset, can fly long distances while foraging. Actually, the phylogenetic component can be very important. Tinamids and Cracids for instance are large-bodied birds, but tend not to fly long distances.

Another important issue is that while some of the species disperse ingested seeds through defecation, others regurgitate the seeds, which impacts the GPT. In the related paper submitted as supporting file for review (Sorensen et al.), it is argued that most birds disperse seeds through defecation, and this should not be a problem, but I strongly disagree. Most toucans, cotingids and trogons, for instance, which are among the large-bodied species in this study, regurgitate the large seeds.

Ln 347. The extinction simulations need to be presented in more detail. What is the condition for a secondary extinction? If a plant only goes extinct when it loses all partners, then the extinction simulations are much more restrictive for topological changes than for functional changes.

Dear reviewers,

We appreciate your thoughtful and constructive feedback on our manuscript, which in our opinion helped to improve the manuscript. Below we provide a point-by-point response to each of your suggestions and explain how we have accounted for them in the revised manuscript. Particularly, we performed additional analyses demonstrating the robustness of our findings. Our responses to the referees' comments are indicated by bold font. To facilitate the review process, we have also highlighted the changes made to the manuscript in red font in the manuscript text. Please note that line numbers in our responses refer to the revised version of the manuscript.

Thank you again for your valuable suggestions on how to improve our Research article,

Isabel Donoso and Matthias Schleuning (on behalf of all authors)

Reviewer #1 (Remarks to the Author):

This paper compares commonly used simulations of coextinction within mutualistic networks to simulations of the loss of ecological function (long-distance dispersal) following simulated extinction. The key conclusions – that functional loss can outpace species loss and that ecological function can be better incorporated in network models – are important and I think articulated well. From one perspective, the paper's primary conclusions mirror basic ideas/predictions present in the literature, and the qualitative conclusions of the model are more or less guaranteed to be found on first principle. Because networks are typically nested and larger animals can eat a broader diversity of fruits, the loss of the largest animals will leave disproportionately many plant species without partners. Because dispersal distances increase with body size, and because body size distributions are roughly log-normally distributed, the loss of the largest animals is essentially guaranteed to cause disproportionate decreases in dispersal distance.

All of that said, the integration of field network data and trait data, as well as trait-based estimates of dispersal distances, allows the authors to get a quantitative estimate of the scope of these simulated downsizing effects if they were to occur in a real community, making this a useful contribution to the field. I also think it is a useful strategy to consider functional components within the context of the commonly used topological coextinction models. Hopefully this makes it easier to put these functional loss ideas in the context of previous literature using such coextinction simulations. So overall I think the conclusions of the model, and the large scope of functional loss demonstrated, provides a useful demonstration applicable to those modeling mutualistic networks and those studying them in the field. I generally found the presentation to be clear. My one major comment points out a methodological question that may quantitatively impact the conclusions.

RESPONSE: Thank you very much for your positive assessment of our manuscript and for stressing the novelty of considering functional components relative to structural (topological) changes in interaction networks. Below we provide information on how we have addressed your comment regarding our methodology.

Major comments

Displacement values are simulated by taking the product of a draw from a GPT distribution and a draw from the FS distribution. This product seems to be how displacement values are typically simulated. What I do not understand is why this product is then used as the mean of a lognormal displacement distribution, with the actual displacement value drawn from that distribution (L 318). Ecologically, I do not understand where this additional variation is coming from. Why I think this is important is because the long tail of the lognormal distribution will make some very high values for displacement and, in turn, dispersal distances. If I understand correctly, these will be disproportionately long for large bodied animals, and this could bias upwards the estimates of functional loss caused by large frugivore loss (which are not measured on the log scale). The authors should avoid this additional step or justify the biological meaning of this additional distribution. I suspect the functional downsizing effects would be less pronounced without this additional step.

RESPONSE: Thank you for this valuable suggestion. We followed your suggestion and now estimate the displacement values without defining an additional displacement distribution. As you suggested, values of seed dispersal distances are now simulated by taking the product of a draw from the GPT distribution and a draw from the FS distribution (see lines 328-340). As you had expected, the absolute values of long-distance seed dispersal were shorter after removing the displacement distribution from the simulations (compare Supplementary Fig. 5 in the revised version to the original manuscript) and are now more coherent with the few empirical studies on LDD (e.g., see Jordano et al. 2007). Most importantly, after this adjustment of our simulation, downsizing effects on LDD remained qualitatively identical despite the changes in the absolute values (new Fig. 3 and Fig. 4 in the main text).

I also suggest clarifying how functional loss is actually calculated. I think it's the 95th percentile of the simulated dispersal distance of whatever frugivores persist versus the 95th percentile of simulated dispersal distance of the full frugivore assemblage before any extinction, but this was unclear in the text.

RESPONSE: This information was previously only provided in the Methods sections (see lines 357-360). We now explain this definition also at the beginning of the Results (see lines 136-137), as well as in the Figure caption 3 (lines 146-148).

Specific comments

L25. How the term "structural stability" is used here is different from how structural stability is typically used, which is in the context of dynamical models (e.g., Rohr et al. 2014 Science). Perhaps robustness is a more appropriate term?

RESPONSE: Thank you for this suggestion. We agree that structural robustness might be a more appropriate term and have changed this terminology throughout the manuscript (e.g. Lines 22, 25, 30, 59, 86, 90, 107, 204, 175 or 251).

L52. Rephrase "size has been described as a crucial mechanism mediating defaunation effects on ecosystem functioning". Size itself is not a mechanism.

RESPONSE: We rephrased the sentence, which now reads “size matching has been described as a crucial mechanism...” (Line 53)

L166. This reads as a very general and strong conclusion, so it is worth being clear that these are simulated extinctions.

RESPONSE: Thank you for this suggestion. This has been clarified in lines 175-177, 195, 205 and 248.

L170. Maybe a word is missing here - perhaps “compensate against secondary plant extinctions”

RESPONSE: The sentence was corrected, and now reads “compensate for secondary plant extinctions...”

L 182. Rephrase “is little limited”

RESPONSE: Changed as suggested.

Reviewer #3 (Remarks to the Author):

The authors use extinction simulations in seed dispersal networks to investigate how the loss of frugivorous birds could impact the structure of the network and seed dispersal services. Although the study is interesting and well-written, it is structured in such a way that the authors are continuously comparing the functional versus structural consequences of frugivore loss. Because they only consider that structure changes when species are lost, it is not surprising that structure will change less than function, especially if they consider plants will only die out when all birds that disperse them are extinct. Actually, assuming secondary extinctions of plants due to the loss of frugivores is not realistic. There is very limited evidence that the loss of frugivores result in local extinction of plants, although it can result in dispersal limitation with consequences for population genetics, population size and structure and spatial distribution. I'm not saying that extinction simulations are not informative, but it seems more reasonable to measure the loss of interactions instead of species loss. If a plant loses all of its dispersers it becomes disconnected from the network, but it is unlikely to go extinct, especially in short timespans. If the authors consider the number or proportion of lost interactions or had some way to evaluate changes in the distribution of interaction weights, for instance, the results on the structural versus functional consequences of species loss could be very different. That said, I think relying on this comparison is not the best way to motivate the paper. Instead of opposing the two consequences it would be more interesting to ask how changes in topology translate into functional consequences.

Another important issue is that some of the simplifying assumptions used to estimate seed dispersal distances, disregard important behavioral and physiological characteristics of the bird species that have an important impact on their role as seed dispersers (see below)

We thank the reviewer for these thoughtful comments on our manuscript. We have added several new points to the manuscript according to your suggestions:

- 1. As suggested, we have incorporated interaction loss as another structural measure of network robustness. These new results reinforce our main conclusion that functional changes are more**

severe than structural changes in response to animal downsizing. We have added the new metric to Figure 1, Figure 2 and Figure 4 of the main text, and set these new findings into context in lines 107, 112-115 in the Results and 194-204 in the Discussion.

2. We fully agree with your notion that secondary plant extinctions are rare events, given the high degree of redundancy in seed-dispersal networks and the little empirical support for such events. Actually, this is part of the motivation for this manuscript as we aim to show that structural measures of network robustness are likely to underestimate the actual functional changes in ecological communities. Despite the limitations of such approaches, coextinction models have been/are very influential in network ecology (e.g., see the seminal paper by Memmott et al. 2004 or a recent study by Bascompte et al. 2019). While we think that analyses of coextinctions can be of high relevance, our findings show that measures of structural robustness are insufficient for describing the functional impacts of species extinctions for ecosystems (see also our revised Discussion in lines 174-177; 194-204; 206-208 and lines 211-218).
3. For clarification about which type of structural measure we are using, we now consistently use the term structural robustness in the revised manuscript (as suggested by reviewer 1).
4. We have done additional analyses considering potential behavioural and physiological differences among birds to account for your methodological concerns which confirm the robustness of our findings (see our detailed responses below).

Detailed comments:

The introduction is a bit tiresome due to excessive repetitions on what is unknown, unclear or little understood.

RESPONSE: We have revised the wording in parts of the Introduction. Nevertheless, we think that it is important to highlight the major gaps of knowledge in the Introduction of the manuscript.

Figure 1. secondary extinctions of plants due to the loss of frugivores seem rather unlikely. The consequences are probably seen in terms of genetic diversity and changes in spatial distribution over time.

RESPONSE: We agree with the referee that secondary extinctions of plants are unlikely and discuss this in the manuscript (see lines 206-208 and lines 211-214). We also state that changes in seed dispersal can lead to changes in the spatial distribution of plants (lines 71-74). As suggested, we have added another measure of structural robustness, i.e., interaction loss in response to frugivore extinction.

Ln 143. This comparison is strongly dependent on how you determine secondary extinctions. Since you assume plants will only undergo extinction after they lose all their partners, then it is expected that structure will be more robust than function.

RESPONSE: As suggested, we have added interaction loss as an alternative way to measure the structural robustness of networks. This additional analysis confirms our main finding that the functional consequences of animal downsizing are more severe than the structural consequences in terms of species and interaction loss (see revised Figures 2-4 in the main text). Actually, structural and functional consequences don't have to differ by definition, but depend on the redundancy among frugivore species in terms of the quantity and quality of the provided seed-dispersal services. Apparently, redundancy is higher in terms of structural robustness compared to

scenarios where the actual functional contributions in terms of seed dispersal distances are considered. This important difference is discussed in the manuscript (see lines 175-177; 194-218).

Ln 164. The discussion also hinges on the structure vs function comparison, which I think is not the best strategy since it depends on how you measure each

RESPONSE: In line with the assessment of reviewer 1 (see above), we think that this comparison is the key novelty of our study and have kept this comparison as the main structuring element of the manuscript. As explained above, this comparison is relevant because studies of structural robustness are very influential in network ecology (see e.g. Schleuning et al 2016; Bascompte et al. 2019). To be more inclusive in terms of how to measure structural changes in networks, we have included interaction loss as an additional metric and discuss the new findings in the revised Discussion (lines 194-204).

Ln 166-169. These results can be obtained from structural network metrics and redundancy metrics. I don't think this is an important finding emerging from your simulations, but an explanation for the obtained results.

RESPONSE: We have rephrased the sentence and we now discuss these findings in a wider context, also including the new findings for interaction loss (194-218).

Ln 199. It is not clear if you are referring to your findings or those of the cited references

RESPONSE: We have clarified this in the revised manuscript (lines 188-190). Here, we refer to our findings, which are later related to the cited references.

Ln 272. This decision needs to be explained in more detail

RESPONSE: We added a few sentences in lines 281-283 to explain this decision in more detail. Since the seed-dispersal models are stochastic and to reduce stochastic noise in the simulations, we simulated more interaction events than had been empirically observed. By multiplication with a constant value, we maintained the relative frequency of interaction links between plant and bird species from the empirical observations.

Ln 281. how good is the fit of this allometric relationship?

RESPONSE: We found strong support for a positive relationship between body mass and gut passage time ($n = 37$ species, $R^2 = 0.69$, $p < 0.001$). We added the statistics to the main text (line 293). Further details are also provided in the electronic supplementary material (Fig. S1) of Sorensen et al. (included with the original submission).

Ln 287-298. Movement distance varies a lot depending on habitat preferences of different bird species. Understory specialists, such as some of the Passerelidae species in the networks tend to fly very short distances whereas canopy birds with similar body mass, such as many Thraupidae in the dataset, can fly long distances while foraging. Actually, the phylogenetic component can be very important. Tinamids and Cracids for instance are large-bodied birds, but tend not to fly long distances.

RESPONSE: We agree with the referee that different bird species differ in their movement distances depending on their behaviour. Importantly, our study does not include ground-dwelling frugivores (such as Tinamids) which have a distinct movement behaviour. Moreover, the networks are dominated by canopy foragers as most crops were actually presented in the forest canopies at the respective study sites. Therefore, we think that the allometric approach is a reasonable approximation of movement distances across the here considered bird species (given the fact that the allometric equations were also derived from a wide range of bird groups with a similar range in body sizes, see Supplementary figure 4). Nevertheless, we agree that Cracids may be behaviourally distinct and more reluctant to fly over long distances compared to other bird taxa. To account for this, we re-run our simulations removing all the Cracids from our eight seed-dispersal networks (i.e. 7 species in total, corresponding to the genera *Penelope*, *Aburria*, *Chamaepetes* and *Ortalis* in Supp. Table 2). As expected, absolute values of long-distance seed dispersal were shorter after excluding these large-bodied species (compare Fig. R1 in this file to Supplementary Figure 5). Most importantly, these additional simulations show that the main findings of our study remained the same and that the consequences of downsizing were more severe for functional than for structural properties of seed-dispersal networks, independent of whether Cracids were included or excluded from the simulations (compare Figs. R2-R4 in this file to the new Figs. 2-4 in the main text). We have not included this additional analysis in the revised manuscript because it represents a rather specific case and the findings are identical in both simulation runs (with vs. without Cracids), but we would be happy to add it to the manuscript if the editor considers this to be important.

Figure R1 | Violin plots representing the seed-dispersal kernel – considering the 0.95 quantile - for each of the eight seed-dispersal networks collected across the Andes when Cracidae were excluded from the simulations. Shown are the distributions of the simulated seed dispersal distances (in meters) for the entire plant community of each of the eight empirical seed-dispersal networks (depicted in different colours). Compare to Supplementary Fig. 5 for the simulations including Cracids.

Figure R2 | Structural changes in seed-dispersal networks in response to simulated animal extinctions when Cracidae were excluded from the simulations. Consequences of size-structured bird extinctions (downsizing; red line) and random bird extinction (mean across 1000 iterations; black line) on (a) the number of interactions and (b) secondary plant species extinctions. Compare to Fig. 2 for the simulations including Cracids.

Figure R3 | Functional changes in seed-dispersal networks in response to simulated animal extinctions when Cracidae were excluded from the simulations. Consequences of size-structured bird extinctions (downsizing; red line) and random bird extinction (mean across 1000 iterations; black line) on long-distance seed dispersal (LDD). Values on the y-axis show the percentage of functional change defined by the 0.95 quantile of the community seed-dispersal distance relative to the value in the original network. Compare to Fig. 3 for the simulations including Cracids.

Figure R4 | Effect size of downsizing on structural and functional losses in seed-dispersal networks when Cracidae were excluded from the simulations. Effect sizes for the number of interactions and secondary plant extinctions (structural losses; yellow and green lines, respectively), as well as for long-distance seed dispersal (functional loss; blue line) along the entire extinction sequence. Compare to Fig. 4 for the simulations including Cracids.

Another important issue is that while some of the species disperse ingested seeds through defecation, others regurgitate the seeds, which impacts the GPT. In the related paper submitted as supporting file for review (Sorensen et al.), it is argued that most birds disperse seeds through defecation, and this should not be a problem, but I strongly disagree. Most toucans, cotingids and trogons, for instance, which are among the large-bodied species in this study, regurgitate the large seeds.

RESPONSE: We agree with the referee that some seeds might be unlikely to pass through the avian gut, but would rather be regurgitated. To account for this difference in seed handling, we set up an alternative simulation run excluding regurgitated seeds. We base this additional simulation on two main points based on the literature. First, the likelihood of regurgitation events depends on the size ratio between a seed and a frugivore (Uriarte et al. 2011; Gaspering & Pizo 2012). Second, the time of regurgitation is shorter than the time of gut passage (Levey 1986; Gasperin & Pizo 2012). Therefore, regurgitation events are very unlikely to contribute to LDD and were therefore excluded from an alternative simulation run. To implement this in the simulation, we fitted a quantile regression between fruit size and bill width across all observed interaction events (across the 8 networks). All observed interaction events above the 0.75 quantile of this relationship were excluded from our simulations assuming that these seeds were too large to pass through the avian gut (corresponding to 25% of all observed interaction events). Most importantly, this additional simulation did not change our main results and confirmed the effects of animal downsizing on structural and functional changes in seed-dispersal networks (Supplementary Fig. 7-9). As we think that this suggestion of reviewer 3 might be of relevance for the readers of this manuscript, we added a description of this additional analysis – along with the corresponding figures - in the new

Supplementary Note 1. We further included the main components and findings of this addition in the main text (lines 297-302).

Ln 347. The extinction simulations need to be presented in more detail. What is the condition for a secondary extinction? If a plant only goes extinct when it loses all partners, then the extinction simulations are much more restrictive for topological changes than for functional changes.

RESPONSE: We have clarified how secondary extinctions and interaction loss (added to the revised manuscript) were calculated (see the figure caption 1 and lines 107-111).

References:

- Jordano, P., Garcia, C., Godoy, J. A. & Garcia-Castano, J. L. Differential contribution of frugivores to complex seed dispersal patterns. *Proc. Natl. Acad. Sci. USA* 104, 3278–3282 (2007).
- Memmott, J., Waser, N. M. & Price, M. V. Tolerance of pollination networks to species extinctions. *Proc. R. Soc. B.* 271, 2605–2611 (2004).
- Bascompte, J., García, M. B., Ortega, R., Rezende, E. L. & Pironon, S. Mutualistic interactions reshuffle the effects of climate change on plants across the tree of life. *Sci. Adv.* 5, eaav2539 (2019).
- Gasperin, G., & Pizo, M. A. Passage time of seeds through the guts of frugivorous birds, a first assessment in Brazil. *Revista Brasileira de Ornitologia*, 20, 48-51(2012).
- Levey, D. J. Methods of seed processing by birds and seed deposition patterns. In *Frugivores and seed dispersal*, 147-158. (Springer, Dordrecht, 1986).
- Schleuning, M. et al. Ecological networks are more sensitive to plant than to animal extinction under climate change. *Nat. Commun.* 7, 13965 (2016).
- Uriarte, M., Anciães, M., Da Silva, M. T., Rubim, P., Johnson, E., & Bruna, E. M. Disentangling the drivers of reduced long-distance seed dispersal by birds in an experimentally fragmented landscape. *Ecology*, 92, 924-937 (2011).

Reviewers' Comments:

Reviewer #1:

Remarks to the Author:

I appreciate the author's revision of the text and additional analyses inspired by both reviews, which I think have made the paper stronger. I share the other reviewer's concerns about the real-world applicability of these sorts of topological coextinction models, and I think the text now better acknowledges some of their shortcomings while improving an aspect that network simulation models have not captured to date (components of ecological function such as long-distance dispersal).

One change I noticed involved the flight speed allometry. The authors now use the 0.17 exponent from Tucker et al. 1973 rather than the 0.15 exponent of Alerstam et al. 2007. Because there is a more recent / commonly used estimate, I'd be curious whether this change impacts the conclusions. I would guess it increases the magnitude of the functional decay effect sizes. If there's much of a difference, it would be worth 1) justifying why one value was chosen over the other, 2) using the more recently calculated exponent, or 3) perhaps even finding more data since the 2007 Alerstam paper and performing an updated analysis.

Reviewer #3:

Remarks to the Author:

The authors have addressed all the concerns raised by the reviewers satisfactorily. The results are convincing and the main message is clear. I consider the manuscript has improved considerably and I have no further suggestions.

Dear reviewers,

We are glad that the new version of the manuscript met your expectations. Below we have clarified the remaining minor concern raised by Reviewer 1. Please note that when mentioning line numbers, we refer to the pdf version with changes accepted (pdf file generated during the submission process).

Thank you again for your valuable suggestions on how to improve our Research article,

Isabel Donoso and Matthias Schleuning (on behalf of all authors)

Reviewer #1 (Remarks to the Author):

I appreciate the author's revision of the text and additional analyses inspired by both reviews, which I think have made the paper stronger. I share the other reviewer's concerns about the real-world applicability of these sorts of topological coextinction models, and I think the text now better acknowledges some of their shortcomings while improving an aspect that network simulation models have not captured to date (components of ecological function such as long-distance dispersal).

RESPONSE: We thank the reviewer for all the comments and suggestions that have helped to improve the quality of the manuscript throughout the review process.

One change I noticed involved the flight speed allometry. The authors now use the 0.17 exponent from Tucker et al. 1973 rather than the 0.15 exponent of Alerstam et al. 2007. Because there is a more recent / commonly used estimate, I'd be curious whether this change impacts the conclusions. I would guess it increases the magnitude of the functional decay effect sizes. If there's much of a difference, it would be worth 1) justifying why one value was chosen over the other, 2) using the more recently calculated exponent, or 3) perhaps even finding more data since the 2007 Alerstam paper and performing an updated analysis.

RESPONSE: The equation developed by Alerstam et al. 2007 is specifically based on migratory species. Therefore, we decided to use the allometric equation from Tucker et al. 1974 because it derived flight speed from avian aerodynamic measures collected during wind tunnel experiments. Therefore, we consider it to be more generally applicable to all types of bird species. We have clarified the use of the equation developed by Tucker in the main text (see lines 273-276).

In any case, we would like to highlight that changing the exponent from 0.15 to 0.17 (the difference between the Alerstam and Tucker equation) does not affect the conclusions of our manuscript. Actually, results from a global sensitivity analysis performed in another study by several of the same authors (Sorensen et al. 2020, currently available as a preprint at <https://doi.org/10.1101/2020.02.23.958454>) showed that the estimates of long-distance dispersal were robust to variation in the parameters of the flight speed equation (see Figure 4 at <https://doi.org/10.1101/2020.02.23.958454>).

Reviewer #3 (Remarks to the Author):

The authors have addressed all the concerns raised by the reviewers satisfactorily. The results are convincing and the main message is clear. I consider the manuscript has improved considerably and I have no further suggestions.

RESPONSE: The reviewer did not have additional comments.